# X-Pruner: An Adaptive Pruning Method with Self-Compensation Driven by Reinforcement Learning for Language Models

## Abstract

As small language models (SLMs) emerge as the backbone of on-device, mobile, and edge devices, their constrained computational and memory budgets necessitate aggressive yet reliable pruning. Compared with their larger counterparts, SLMs exhibit more sensitivity to parameter removal, rendering the design of robust pruning strategies particularly challenging. Existing post-training pruning techniques, predominantly designed for large language models (LLMs), rely on static criteria computed from tiny calibration sets, often resulting in suboptimal generalization. In this paper, we present X-Pruner, an unstructured adaptive pruning framework featuring a variable-exponent importance metric. To unlock its full potential, we introduce a reinforcement learning-based search algorithm that efficiently identifies optimal parameter configurations. We further reveal that the pruning path itself influences post-pruning performance and creatively propose the self-compensation mechanism, which rectifies pruning-induced errors through layer-wise adaptive adjustments; grounded in this insight, we also devise a unified path-scoring function to evaluate and select optimal pruning sequences across diverse target models. Extensive experiments on multiple language benchmarks demonstrate that X-Pruner consistently surpasses state-of-the-art post-training pruning techniques under comparable settings—achieving superior performance without any retraining—and in certain cases, even outperforms approaches involving update weights.

## 1 Introduction

Large Language Models (LLMs) Touvron et al. (2023); Abdin et al. (2024); OpenAI et al. (2024) have revolutionized the field of natural language processing (NLP), demonstrating exceptional capabilities across diverse tasks such as language understanding, generation, and reasoning (Bommarito & Katz, 2022; Wei et al., 2022; Bubeck et al., 2023). However, exponential growth in their parameter size, which often extends to billions of dimensions, poses substantial deployment obstacles, particularly in resource-constrained scenarios, including edge computing systems and mobile applications (Zheng et al., 2025; Girija et al., 2025). In this context, Pruning LeCun et al. (1989); Hassibi et al. (1993); Han et al. (2015) has emerged as a fundamental optimization strategy, offering a methodology to eliminate superfluous parameters while preserving the efficacy of the model. However, a fundamental limitation persists in the poor cross-scale generalization of the existing pruning methodologies, which are primarily developed for LLMs but exhibit significantly degraded performance when applied to small language models (SLMs).

Conventional pruning techniques frequently fail to preserve satisfactory performance levels in SLMs — a deficiency attributable to their underlying design. Most established post-training pruning approaches, including contemporary methods, employ static metrics that demonstrate high sensitivity to variations in both the data and the model (Du et al., 2023). This sensitivity is further amplified in compact models. For instance, Wanda Sun et al. (2024) utilizes first-order information, while Pruner-Zero Dong et al. (2024) employs genetic algorithms to automate the search for pruning metrics but ultimately converges to a fixed, predetermined metric. Such inflexible metrics lack the adaptability required for robust performance across diverse model architectures and data conditions. This methodological constraint proves particularly detrimental for SLMs, where the impact of pa-

rameter removal is substantially more pronounced, as shown in Figure 1. The smaller the scale of the model, the more pronounced the impact of pruning becomes. The static nature of these pruning criteria fails to provide theoretical guarantees or empirical consistency in maintaining baseline performance. Such sensitivity mainly reflects redundancy differences; deeper causes are analyzed later.

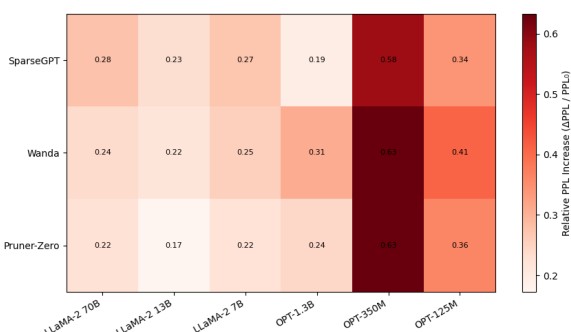

Figure 1: Relative perplexity (ppl) increase of different pruning methods at 50% sparsity. The darker the colour, the greater the decline in performance.

This gap highlights the urgent need to advance pruning research specifically tailored for SLMs. The primary motivation for pruning SLMs stems from their frequent deployment in resource-constrained environments, where even marginal reductions in model size can yield substantial improvements in computational efficiency and accessibility.

To address these limitations, we propose X-Pruner, a layer-wise adaptive pruning framework that jointly optimizes both pruning criteria and pruning path to enhance post-pruning performance, particularly for SLMs. As illustrated in the left half of Figure 2, each transformer layer undergoes pruning based on a parameterized importance score formulated as $W^x \times G^y$, where exponents $(x, y)$ are adaptively searched via a reinforcement learning agent. This agent, depicted in the Figure 2's upper right part, explores the parameter space in a three-phase process: random exploration, strategy search, and local refinement—guided by perplexity feedback to identify optimal exponent configurations without retraining. Beyond this, X-Pruner introduces a novel insight: the pruning path significantly impacts final model quality due to a self-compensating effect among layers, which implies that pruning at each layer inherently incorporates an offset or elimination of cumulative errors introduced by preceding-layer pruning. $\Delta_{\text{margin}}$ in the figure above represents the marginal cost, which is defined as the net increase in loss induced by, under the current pruned state. A precise definition and detailed explanation of this concept will be provided in later sections. To quantify the performance of different paths, we develop a unified path scoring function that balances local pruning cost and compensation potential, thereby enabling the selection of the globally optimal pruning path. This fine-grained unified optimization of 'what to prune' and 'when to prune' endows X-Pruner with superior scalability and adaptability across diverse model architectures and data conditions. Notably, Pruning SLMs is, in a sense, analogous to pruning LLMs at a much higher sparsity rate. This suggests that our method not only advances SLM pruning but also provides a promising framework for pursuing high sparsity regimes in LLMs. Our principal contributions include: (i) **Adaptive pruning metric**: We propose a variable-exponent weight-gradient formulation, offering greater flexibility to trigger compensation effects; (ii) **Error compensation mechanism**: We exploit an inherent compensation mechanism, where pruning actions spontaneously offset part of the accumulated error; (iii) **Fine-grained RL controller**: Our reinforcement learning framework locally maximizes compensation at each layer without retraining; (iv) **Optimized pruning path**: A unified scoring strategy selects paths that globally maximize compensation via inter-layer coupling.

Further clarification is needed regarding the interplay of these components. The adaptive pruning metric serves as the framework's core, while the error compensation mechanism provides the theoretical basis for its superior performance. The RL-based search ensures local optimality at the layer level, and the pruning path achieves global performance optimization.

## 2 METHOD

In this section, we provide a detailed explanation of the components and design principles of the X-Pruner framework, including the adaptive pruning metric (Section 2.1), the error propagation and compensation mechanism (Section 2.2), the reinforcement learning-based search algorithm (Section 2.3), and the layer-wise optimization of the pruning path (Section 2.4).

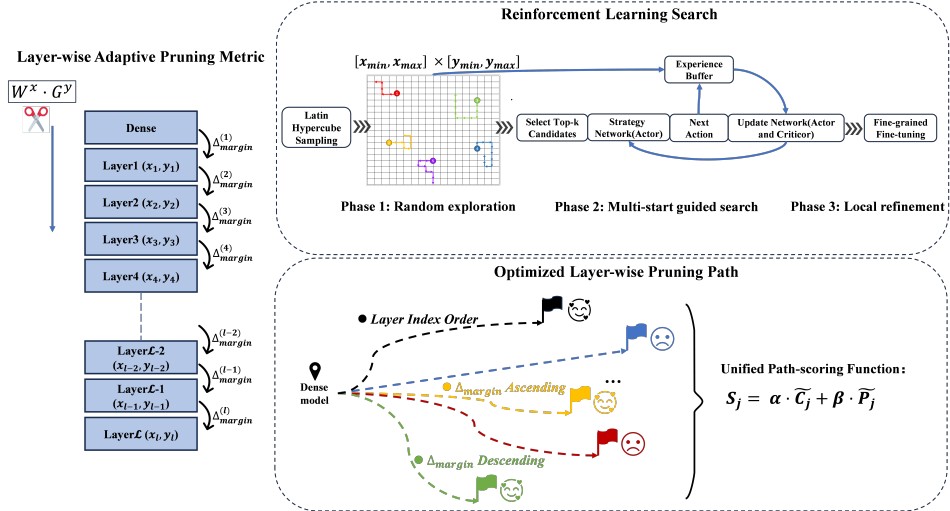

Figure 2: Overview of X-Pruner framework. The framework integrates a flexible pruning metric $W^x \cdot G^y$, an inherent error compensation mechanism (manifested as $\Delta_{\text{margin}}$), a RL-based exponent search and a globally optimized pruning path selected via a unified scoring function.

## 2.1 ADAPTIVE PRUNING METRIC

Consider a model with $\mathcal{L}$ layers (transformer blocks). For the $k$th layer, $W_k \in \mathbb{R}^{d_{out} \times d_{in}}$ represents the original weight matrix and $G_k$ is its corresponding gradient matrix. We define a new scoring matrix as:

$$S_k = W_k^{x_k} \circ G_k^{y_k} \tag{1}$$

where $x_k, y_k$ are adjustable power exponents; $\circ$ represents the Hadamard product (element-by-element multiplication). All operations are performed element by element. Based on the scoring matrix $S_k$, a gating matrix $M_k \in \{0, 1\}^{d_{out} \times d_{in}}$ is constructed by retaining the top $(1 - r)$ proportion of elements (where $r$ is the pruning rate). The pruned weight matrix is obtained by element-wise multiplication of $M_k$ and the original weight matrix $W_k$. The single-layer pruning error is defined as the difference between the pruned and original weights, which equals the element-wise product of $(M_k - 1)$ and $W_k$.

## 2.2 ERROR PROPAGATION AND COMPENSATION MECHANISM

We will analyze the error propagation and compensation mechanism in the layer-by-layer pruning process through a rigorous mathematical framework.

**Forward Error Propagation.** When weights in layer $k$ are pruned, the change can be modeled as a small perturbation $\Delta W_k$ to the original weight matrix $W_k$. This perturbation interacts with the layer input $h_k \in \mathbb{R}^d$, modifying the pre-activation $W_k h_k + b_k$, where $b_k$ is the bias vector. The resulting output error can be approximated using a first-order Taylor expansion as

$$\Delta h_{k+1} \approx J_f(W_k h_k + b_k) \cdot \Delta W_k \cdot h_k, \tag{2}$$

where $J_f(\cdot) \in \mathbb{R}^{d \times d}$ denotes the Jacobian matrix of the activation function evaluated at the pre-activation point. This formulation captures how the pruning-induced perturbation $\Delta W_k \cdot h_k$ is further transformed by the local slope of the activation, resulting in a variation in the output. The expression reveals that pruning errors propagate anisotropically, as they are jointly influenced by the direction and magnitude of the input $h_k$ and by the local sensitivity of the activation encoded in $J_f$.

**First- and Second-Order Expansion of the Total Loss Function.** The foundational work, known as Optimal Brain Damage LeCun et al. (1989), introduced a second-order pruning framework that approximates loss change using only the diagonal of the Hessian for efficiency. Later, Optimal Brain Surgeon Hassibi et al. (1993) improved upon this by incorporating off-diagonal terms for greater accuracy. Although we refer to these two papers to obtain Formula 3, the Hessian matrix was only

used for mathematical derivation in our work. By incorporating pruning-induced perturbations, the total loss function can be approximated as:

$$\ell \approx \ell_0 + \sum_{k=1}^{i} \frac{\partial \ell}{\partial W_k} : \Delta W_k + \frac{1}{2} \sum_{k,m=1}^{i} \Delta W_k : \frac{\partial^2 \ell}{\partial W_k \partial W_m} : \Delta W_m \tag{3}$$

Here, the symbol ":" denotes the Frobenius inner product (tensor contraction). The first term $\ell_0$ represents the baseline loss. The second term captures the **first-order linear contribution** from pruning-induced perturbations and local gradients. The final term reflects the **second-order contribution** under the influence of the Hessian matrix, that is, the second-order partial derivatives of the loss with respect to the weights in the formula.

**Compensation Mechanism.** In order to maximize the absorption of existing errors at this level, the goal is to construct a compensatory mask $M_{i+1}$ at layer $i + 1$ by selecting a suitable data point $(x_{i+1}, y_{i+1})$. We then aim to minimize the propagated error through the following formula:

$$\frac{d\mathcal{L}}{d(x_{i+1}, y_{i+1})} = \frac{\partial \mathcal{L}}{\partial W_{i+1}^{\text{pruned}}} \cdot \frac{\partial M_{i+1}}{\partial (x_{i+1}, y_{i+1})} \circ W_{i+1} \tag{4}$$

The two derivatives on the right represent, respectively, the residual error propagated from the upper layers and the "movement direction" in the $x, y$ space that influences the pruning mask. The ingenuity of formula 4 lies in mapping the pruning error to the $(x, y)$ space, which gives us more freedom to handle error. As long as $(x, y)$ is not a constant input, we can use the gradient direction to adjust pruning and cancel out the residual error from upper layers (i.e., make the inner product tend to zero).

Suppose that there exists a layer $j$ that can compensate for the error induced by pruning at a previous layer $k$, the second-order compensation can be expressed as:

$$\Delta W_j^{\text{comp}} = - \left[ \frac{\partial^2 \mathcal{L}}{\partial W_j^2} \right]^{-1} \cdot \frac{\partial^2 \mathcal{L}}{\partial W_j \partial W_k} \cdot \Delta W_k \tag{5}$$

If we explicitly solve the Hessian and accurately update layer $j$ along the above equation, we can completely offset the pruning error of layer $k$. But this is challenging for models with hundreds of billions of parameters. We have customized a reinforcement learning search algorithm as an alternative to the Hessian matrix.

## 2.3 REINFORCEMENT LEARNING SEARCH ALGORITHM

While prior work (Zhang et al., 2022a) also employed RL for pruning, their controller operated at a coarse layer-level granularity. In contrast, our approach leverages a fine-grained RL controller to adaptively optimize the pruning metric itself ($x$, $y$ exponents), thereby directly influencing intra-layer sensitivity.

The proposed RL algorithm efficiently searches for optimal pruning parameters per layer through online policy optimization, thereby avoiding the computational cost associated with Hessian-based methods. Notably, pruning and searching occur simultaneously without requiring the training of a separate policy network.

As shown in Algorithm 1, it proceeds in three phases: (1) **Exploration** uses Latin Hypercube Sampling and policy/value networks (actor-critic) to evaluate diverse candidates, storing experiences in a replay buffer; (2) **Exploitation** refines top candidates using $\epsilon$-greedy search with simulated annealing to avoid local minima; (3) **Fine-tuning** reduces step size to converge precisely. This structured process reduces computational overhead while maintaining accuracy in selecting effective pruning parameters.

## 2.4 OPTIMIZATION OF PRUNING PATH

The pruning path—that is, the order in which layers are pruned—plays a critical role in determining the model's final performance. This is primarily due to the cumulative nature of error propagation and the existence of a layer-wise compensation mechanism.

---

**Algorithm 1** RL Search Algorithm for Layer-wise Pruning.

---

**Input:** Pruning layer $k$, sparsity $\rho$, search range $x \in [x_{\min}, x_{\max}]$, $y \in [y_{\min}, y_{\max}]$
**Parameter:** Step size $\delta$, exploration steps $N_e$, exploitation steps $N_g$, refinement steps $N_r$
**Output:** Optimal pruning parameters $(x^*, y^*)$, minimizing perplexity (PPL)

  1: Define basic action space $\mathcal{A} = \{$up, down, left, right$\}$
  2: Initialize Strategic Network: actor $\pi$, Value Network: critic $v$, Replay Buffer $\mathcal{B}$
  3: Generate initial search starting points set $S$ via Latin Hypercube Sampling
  4: **for all** exploration point $(x, y)$ in $S$ **do**
  5:     **for** $t = 1$ to $N_e$ **do**
  6:         Select random or $\pi$-guided action
  7:         Evaluate new parameters; store experience in $\mathcal{B}$
  8:         Periodically update $\pi$, $v$ from $\mathcal{B}$
  9:     **end for**
10: **end for**
11: Select top-performing points as candidate set $Q$
12: **for all** $(x, y) \in Q$ **do**
13:     **for** $t = 1$ to $N_g$ **do**
14:         Perform $\epsilon$-greedy and simulated annealing guided search
15:         Evaluate, store in $\mathcal{B}$, and periodically update $\pi$, $v$
16:     **end for**
17: **end for**
18: Fine-grained local refinement around best $(x^*, y^*)$ for $N_r$ steps
19: **return** optimal parameters $(x^*, y^*)$

---

**Importance of Pruning Path.** Downstream layers can correct upstream pruning errors, but pruning them too early disrupts this compensation, causing irreversible degradation. Second-order analysis reveals that error correction depends on cross-layer dependencies. Hence, pruning should consider not just what to remove, but when, preserving compensation capacity to maintain performance under high sparsity.

**How to Determine the Optimal Pruning Path.** Define the pruning sequence as a permutation $\pi = (\pi_1, \ldots, \pi_L)$. For each pruning step, in the context of considering path effects, rewrite formula 3:

$$\ell(\pi) = \ell_0 + \underbrace{\sum_{t=1}^{L} \left\langle \nabla_{\pi_t}^{(\pi_{<t})}, \Delta W_{\pi_t} \right\rangle}_{\text{First-order local cost}} + \underbrace{\tfrac{1}{2} \sum_{s<t} \Delta W_{\pi_s} : H_{\pi_s \pi_t} : \Delta W_{\pi_t}}_{\text{Second-order cross-layer compensation}} \quad (6)$$

The first-order term captures the immediate impact of pruning on the loss through the conditioned gradient $\nabla_{\pi_t}^{(\pi_{<t})}$, which represents the gradient of the loss with respect to the weights of layer $\pi_t$ under the influence of previously pruned layers $\pi_{<t}$. The second-order term reflects the compensatory interactions between layers via the conditioned Hessian matrix $H_{\pi_s \pi_t}$, encoding how the perturbation in layer $\pi_s$ influences the loss sensitivity to changes in layer $\pi_t$. This formulation leads to two fundamental principles for pruning path: (i) **prune layers with minimal local cost first**, and (ii) **preserve layers with high compensatory potential until the end**.

By adhering to these principles, we can minimize the cumulative pruning loss $\ell(\pi)$ along the pruning path. The ideal strategy first removes layers that incur negligible loss while preserving the flexibility of more influential layers. This sequential approach yields a smoother perplexity curve and improved final performance.

**Unified Scoring Function for Pruning Path.** To determine the optimal pruning order, we propose a unified scoring function that integrates three essential factors: local pruning cost, interlayer compensation capability, and layer depth. $L$ layers are indexed by $j \in \{0, 1, \ldots, L-1\}$. For each layer $j$, we define: (i) $\text{Cost}_j$, the local performance cost caused by pruning the current layer $j$; (ii) $\text{Comp}_j$, the compensation ability of layer $j$ for prior layers' pruning errors; (iii) $\text{Reach}(j) = \dfrac{L - j}{L - 1}$, the relative remaining depth of layer $j$; and (iv) $\Delta_{\text{margin}}^{(S)}(j) = \text{PPL}(S \cup \{j\}) - \text{PPL}(S) \approx \text{Cost}_j - \text{Comp}_j$, the

marginal perplexity increase when pruning layer $j$ conditioned on the already pruned set $S$, which reflects the joint contribution of the layer-wise local cost and the compensatory effect.

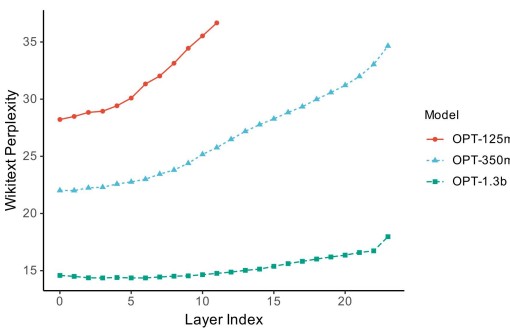

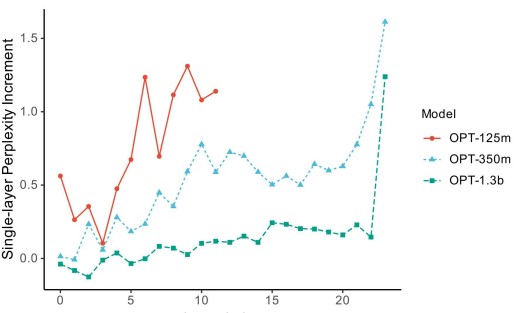

Figure 3: (Top) Perplexity trend during layer-by-layer search. (Bottom) Layer-wise distribution of marginal cost, where the vertical axis represents the net perplexity increment of each individual layer, which corresponds to its marginal cost.

Layer depth influences both local cost propagation and the reach of compensation. We thus re-weight the raw cost and compensation as $\text{Reach}(j) \cdot \text{Cost}_j$ and $\text{Reach}(j) \cdot \text{Comp}_j$, respectively. To ensure comparability across layers, we normalize both the local cost and compensation scores by dividing them by their respective maximum values across all layers. The normalized results, $\widetilde{C}_j$ and $\widetilde{P}_j$, are scaled to a common range for use in the unified scoring function. We compute a linear combination of the normalized components:

$$S_j = \alpha \cdot \widetilde{C}_j + \beta \cdot \widetilde{P}_j \qquad (7)$$

where $\alpha$ and $\beta$ control the relative importance of local cost and compensation ability. We sort layers in ascending order of $S_j$ and prune accordingly.

Based on our analysis and empirical validation, Formula 7 typically leads to three distinct pruning paths across different models in most cases: **(i) Layer-Index Sweep (Position factor dominant).** For the shallow model like OPT-125m illustrated by the red curves in Figures 3, the compensation window is short and interlayer coupling is weak; consequently, pruning in layer-index order preserves the original compensation chain and is therefore the safest strategy. **(ii) $\Delta_{\text{margin}}$ Ascending (Cost factor dominant).** For models like OPT-350m, where the compensation chain is sufficiently long and relatively uniform, pruning layers in ascending order of marginal cost resembles a greedy strategy, in which each step selects the least costly option, leading to minimal overall loss. **(iii) $\Delta_{\text{margin}}$ Descending (Compensation factor dominant).** As shown by the green curves in Figures 3, models like OPT-1.3b exhibit strong compensation potential and pronounced heavy-tail phenomenon Lu et al. (2024), where a small number of tail-end layers accounts for the majority of the error. In such cases, it is preferable to preserve as many compensation-capable layers as possible to mitigate the impact of pruning these high-error layers. The rigorous mathematical derivation, which illustrates how the three general pruning paths can be obtained, follows.

## 2.5 MATHEMATICAL JUSTIFICATION OF ORDER RATIONALE

**I: Layer-index sweep.** Assuming that $\text{Comp}_j \approx 0$, $\text{Cost}_j$ varies slowly and correlates with depth, and $\beta \to 0$, the score satisfies $S_j \propto \widetilde{C}_j$, which is approximately monotone in the layer index $j$. Consequently, the order reduces to

$$S_{j_1} < S_{j_2} \iff j_1 < j_2. \qquad (8)$$

Thus, sorting $S_j$ in ascending order corresponds to **layer-index ascending**.

**II: $\Delta_{\text{margin}}$ ascending.** If the layer-wise compensation scores are nearly constant ($\text{Comp}_j \approx \text{const}$) and their variance is negligible compared to that of the costs ($\text{Var}(\text{Cost}_j) \gg \text{Var}(\text{Comp}_j)$), then we can set $\beta \approx 0$, leading to $S_j \approx \alpha \widetilde{C}_j$. In this case, $S_j$ is proportional to $\widetilde{\Delta}_{\text{margin}}(j) \approx \text{Cost}_j - \text{const}$, so the ordering becomes

$$S_{j_1} < S_{j_2} \iff \Delta_{\text{margin}}(j_1) < \Delta_{\text{margin}}(j_2). \qquad (9)$$

Hence, sorting $S_j$ ascending appears as $\Delta_{\text{margin}}$ **ascending**.

Table 1: Perplexity results on WikiText2 of unstructured 50% sparse models. Our X-Pruner outperforms Wanda and Pruner-Zero. For perplexity, the lower the better.

| Method | OPT | | | Pythia | | | | Qwen2 | | Qwen3 | | Llama3.2 |
|---|---|---|---|---|---|---|---|---|---|---|---|---|
| | 125m | 350m | 1.3B | 160m | 410m | 1B | 1.4B | 0.5B | 1.5B | 0.6B | 1.7B | 1B |
| Dense | 27.66 | 22.00 | 14.62 | 26.88 | 16.31 | 13.16 | 11.79 | 17.61 | 12.37 | 28.63 | 21.60 | 12.96 |
| Magnitude | 7e3 | 6e3 | 1e4 | 6.9e3 | 1.2e3 | 4.9e2 | 3.1e2 | 1.6e2 | 31.79 | 1.1e3 | 3.4e2 | 1.5e3 |
| Wanda | 38.96 | 35.92 | 19.12 | 275.0 | 74.37 | 52.03 | 22.41 | 30.03 | 17.96 | 50.48 | 29.28 | 30.88 |
| Pruner-Zero | 37.69 | 35.91 | 18.19 | 239.5 | 61.06 | 36.63 | 20.15 | 28.42 | 16.44 | 49.14 | 27.25 | 30.04 |
| X-Pruner | **36.67** | **34.25** | **17.80** | **168.3** | **53.24** | **26.87** | **19.20** | **26.44** | **16.01** | **45.63** | **26.47** | **26.31** |

**III: $\Delta_{\mathrm{margin}}$ descending.** When a small tail subset of layers exhibits significantly larger $\Delta_{\mathrm{margin}}$ values (heavy tail) and the compensation term $\widetilde{P}_j$ consistently dominates the unified score despite shrinking over pruning ($\beta > \alpha$), the unified score is effectively dominated by compensation, $S_j \propto \mathrm{Comp}_j$. Since $\Delta_{\mathrm{margin}}(j) \approx \mathrm{Cost}_j - \mathrm{Comp}_j$, fixing $\mathrm{Cost}_j$ yields the relation

$$S_{j_1} < S_{j_2} \iff \mathrm{Comp}_{j_1} < \mathrm{Comp}_{j_2} \iff \Delta_{\mathrm{margin}}(j_1) > \Delta_{\mathrm{margin}}(j_2). \tag{10}$$

Thus, sorting $S_j$ ascending manifests as $\Delta_{\mathrm{margin}}$ **descending**.

## 3 EXPERIMENT

### 3.1 EXPERIMENT SETUP

**Models and Evaluation.** We assess X-Pruner using several prominent families of SMLs: OPT 125m/350m/1.3B Zhang et al. (2022b) and Pythia 160m/410m/1B/1.4B Biderman et al. (2023). To further explore the generalizability of X-Pruner, we apply the developed adaptive pruning metric to some LLM families, such as Qwen2 0.5B/1.5B Yang et al. (2024), Qwen3 0.6B/1.7B Yang et al. (2025), and Llama3.2 1B Grattafiori et al. (2024). The effectiveness of the pruned models is evaluated based on their language modeling performance. Following prior work Xiao et al. (2024); Frantar & Alistarh (2023), we use WikiText validation perplexity (Merity et al., 2016).

Table 2: Per-layer perplexity (PPL) and margin cost ($\Delta$) under three pruning paths.

| Path(i) | | | Path(ii) | | | Path(iii) | | |
|---|---|---|---|---|---|---|---|---|
| **Layer** | **PPL** | $\Delta$ | **Layer** | **PPL** | $\Delta$ | **Layer** | **PPL** | $\Delta$ |
| dense | 14.62 | - | dense | 14.62 | - | dense | 14.62 | - |
| 0 | 14.58 | -0.03 | 2 | 14.40 | -0.21 | 23 | 15.48 | +0.86 |
| 1 | 14.49 | -0.08 | 1 | 14.30 | -0.09 | 15 | 15.64 | +0.16 |
| 2 | 14.37 | -0.12 | 4 | 14.31 | +0.00 | 11 | 15.86 | +0.24 |
| 3 | 14.36 | -0.01 | 5 | 14.25 | -0.05 | 21 | 16.12 | +0.23 |
| 4 | 14.40 | +0.03 | 3 | 14.25 | -0.00 | 17 | 16.26 | +0.14 |
| 5 | 14.36 | -0.03 | 6 | 14.24 | -0.00 | 16 | 16.45 | +0.18 |
| 6 | 14.36 | -0.00 | 9 | 14.25 | +0.01 | 19 | 16.58 | +0.12 |
| 7 | 14.44 | +0.08 | 4 | 14.29 | +0.03 | 20 | 16.82 | +0.24 |
| 8 | 14.51 | +0.07 | 8 | 14.36 | +0.06 | 13 | 16.91 | +0.08 |
| 9 | 14.54 | +0.02 | 7 | 14.47 | +0.11 | 22 | 17.16 | +0.25 |
| 10 | 14.64 | +0.10 | 10 | 14.45 | +0.01 | 11 | 17.22 | +0.06 |
| 11 | 14.76 | +0.11 | 14 | 14.66 | +0.08 | 12 | 17.31 | +0.09 |
| 12 | 14.87 | +0.11 | 13 | 14.77 | +0.11 | 14 | 17.50 | +0.19 |
| 13 | 15.02 | +0.15 | 11 | 14.93 | +0.16 | 10 | 17.64 | +0.14 |
| 14 | 15.13 | +0.10 | 22 | 15.11 | +0.17 | 7 | 17.68 | +0.03 |
| 15 | 15.37 | +0.24 | 13 | 15.30 | +0.18 | 8 | 17.74 | +0.05 |
| 16 | 15.61 | +0.23 | 20 | 15.43 | +0.13 | 4 | 17.80 | +0.05 |
| 17 | 15.81 | +0.20 | 19 | 15.54 | +0.11 | 9 | 17.83 | +0.03 |
| 18 | 16.01 | +0.20 | 18 | 15.70 | +0.06 | 6 | 17.78 | -0.04 |
| 19 | 16.16 | +0.20 | 15 | 15.92 | +0.21 | 3 | 17.82 | +0.04 |
| 20 | 16.35 | +0.16 | 17 | 16.03 | +0.11 | 18 | 17.85 | +0.02 |
| 21 | 16.58 | +0.22 | 21 | 16.37 | +0.23 | 0 | 17.86 | +0.06 |
| 22 | 16.73 | +0.14 | 16 | 16.73 | +0.41 | 1 | 17.85 | -0.01 |
| 23 | 17.97 | +1.23 | 23 | 18.12 | +1.33 | 2 | 17.80 | -0.09 |

**Baselines.** We benchmark X-Pruner against four existing pruning methods, including magnitude pruning Han et al. (2015), SparseGPT Frantar & Alistarh (2023), Wanda Sun et al. (2024), and Pruner-Zero Dong et al. (2024). All of them rely on calibration data to estimate input statistics. To ensure a fair comparison, we use the same calibration dataset as SparseGPT and Wanda—128 sequences with a fixed context length, drawn from the C4 training set Raffel et al. (2023) to estimate input statistics, such as Gradients (G).

**Sparsity.** We adopt a uniform sparsity level across all linear layers under unstructured pruning, following the methodology of Wanda and Pruner-Zero. Our experiments focus on linear layers, as unstructured pruning offers finer granularity and flexibility, enabling effective compression with minimal performance loss—critical for SLM deployment in resource-constrained environments. All implementations build on Wanda's codebase to ensure comparability across pruning approaches.

**Search Settings.** Online RL framework searches for optimal pruning-score exponents $(x, y) \in [0.5, 2.5]^2$ for each transformer layer. The action space adjusts $x$ or $y$ by $\pm 0.1$, and the policy is parameterized by a two-layer MLP. Actor and critic networks are respectively trained using Adam

with learning rates $3 \times 10^{-4}$ and $1 \times 10^{-3}$. Additional implementation details are provided in Appendix B.

## 3.2 EXPERIMENT RESULTS

**Perplexity Advantage.** As shown in Table 1, X-Pruner consistently achieves the lowest perplexity across nearly all model scales and families under the same 50% unstructured sparsity without weight update. Our method significantly outperforms fixed-metric baselines with the maximum observed improvement reaching 30%, which strongly validates the efficacy of our adaptive pruning criterion in maintaining model expressiveness even under aggressive compression.

Table 3: Comparison with the SparseGPT method in terms of mean and standard deviation of perplexity. The values in the table represent the mean ± standard.

| Method | OPT | Pythia | Others |
|---|---|---|---|
| SparseGPT | 29.76±9.59 | 114.1±148.4 | 29.76±112.1 |
| X-Pruner | **29.57±8.36** | **66.90±59.89** | **28.52±91.69** |

**No weight update.** SparseGPT alleviates pruning-induced degradation through weight updates. In contrast, X-Pruner operates under a strictly no-tuning regime. Remarkably, even without weight updates, X-Pruner achieves perplexity comparable to SparseGPT (with updates) on OPT and significantly outperforms it on Pythia (Table 3). This highlights the effectiveness of our metric–path co-design in offsetting the lack of fine-tuning.

**Stability.** In addition to improved performance, X-Pruner also shows superior robustness. As reported in Table 3, the standard deviation of perplexity across checkpoints is markedly lower than that of SparseGPT and the static-metric baselines. This indicates that our framework maintains consistent effectiveness across different model sizes and architectures, thereby offering a more reliable solution in practical deployment scenarios.

## 3.3 EFFECTIVENESS OF OPTIMIZED PRUNING PATH

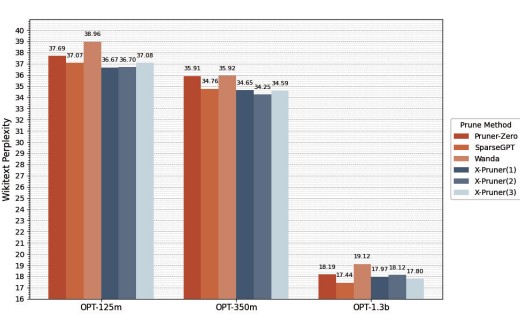

Figure 4: Comparison of three simplified layer pruning paths: (i) layer-index sweep; (ii) $\Delta_{\mathrm{margin}}$ ascending; (iii) $\Delta_{\mathrm{margin}}$ descending.

Under the same no-weight-update setting, the incorporation of dynamic scoring metric and search algorithm enables our method to consistently outperform static approaches (e.g., Wanda and Pruner-Zero), as reflected by the blue bars being overall lower than the red bars in Figure 4. With optimal pruning-path strategy, we further depress post-pruning perplexity, reaching levels comparable to SparseGPT— which performs weight fine-tuning—and even surpassing it on OPT-125m and OPT-350m.

In Table 2, we report the layer-wise perplexity (PPL) and marginal cost $\Delta$ under the three pruning paths for OPT-1.3B. The symbols "–" and "+" mark the sign of $\Delta$: a negative value indicates that the layer's compensatory effect outweighs local-error propagation, thereby reducing overall model perplexity. All three paths exhibit compensation to varying degrees. We argue that such compensatory capacity is an inherent property of the model architecture; static criteria tend to suppress it, whereas dynamic metrics can elicit it, and a well-chosen pruning path amplifies it. This is the fundamental reason for the success of the X-Pruner framework.

## 3.4 ABLATION STUDY

To quantify the contribution of each design choice in X-Pruner, we conduct a series of controlled ablation experiments on OPT-125m under the default 50 % unstructured sparsity. Table 4 reports validation perplexity. Additional experimental results can be found in the Appendix C.

Table 4: Ablation study of key components in X-Pruner.

| Variant | Adaptive metric | RL search | Comp-aware order | ppl $\downarrow$ | $\Delta$ppl |
|---|---|---|---|---|---|
| **Full X-Pruner** | ✓ | ✓ | ✓ | **36.67** | – |
| (a) Fixed exponent | ✗ | ✓ | ✓ | 37.44 | +0.77 |
| (b) Grid/random search | ✓ | ✗ | ✓ | 36.82 | +0.15 |
| (c) Random layer order | ✓ | ✓ | ✗ | 37.16 | +0.49 |

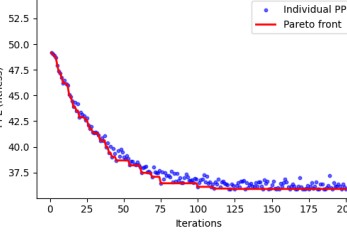 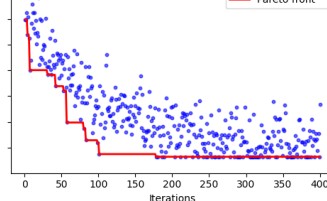 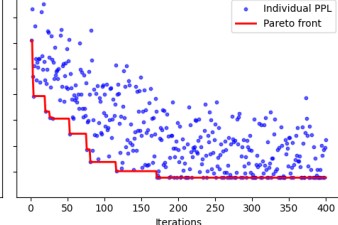

Figure 5: Comparison between RL Search, Grid and Random Search Processes. Individual PPL denotes the individual solution obtained at each iteration. The pareto front represents the curve of historically best results within the current iteration (i.e., the "optimal convergence trajectory").

**Adaptive exponent vs. Fixed exponent.** We conducted extensive random generation tests with fixed exponents (i.e., pruning metric remained unchanged across all transformer layers). Experimental results indicate that the optimal fixed points lie near $x = 1.6$ and $y = 1$, although they exhibit a certain performance gap compared to the dynamic exponents.

**RL Search vs. Grid/Random search.** In terms of both the final convergence outcome and convergence speed, RL search comprehensively outperforms other search methods. As can be readily observed from Figure 5, RL search achieves comparable pruning performance to Grid/Random search with only half the number of iterations, and even lower perplexity. Upon closer inspection, the individual PPL values in RL search remain consistently close to the Pareto front, suggesting that most iterations contribute to progressive optimization rather than random fluctuation. This reflects the stability and efficiency of the RL-based search process.

**Comp-Aware Order vs. Random layer order.** Among the pruning paths generated through multiple random sequences, even the best-performing random path fails to match the optimal path identified by our unified scoring function.

## 4 CONCLUSION

We propose X-Pruner, a complete adaptive pruning framework for small language models (SLMs), featuring a parameterized adaptive importance metric, a reinforcement learning-based search algorithm, and a unified pruning path scoring function. We also reveal a layer-wise error compensation mechanism inherent in the pruning process. X-Pruner addresses the limitations of existing Large Language Models (LLMs) pruning methods, which often suffer from weak adaptability and poor generalization when applied to smaller models. Experimental results show that, under comparable conditions, our method consistently outperforms state-of-the-art pruning techniques. Although X-Pruner is designed with SLMs in mind, it retains the potential to generalize to LLMs. Particularly, in light of the performance degradation commonly observed in LLMs pruning under high sparsity regimes, our strategy of pruning compact models as a means to achieve extreme sparsity offers a novel perspective.

## ETHICS STATEMENT

Our X-Pruner framework provides a powerful tool for adaptively pruning small and large language models, enabling substantial efficiency gains and facilitating deployment in resource-constrained environments. While such compression techniques can democratize access to language technologies, they also pose potential ethical risks. For instance, overly aggressive pruning or misuse of pruning strategies may degrade model reliability, amplify hidden biases, or compromise safety in downstream applications. Moreover, the ability to prune models without retraining could inadvertently be exploited to deploy lightweight but insufficiently validated systems in sensitive domains. We therefore urge researchers and practitioners to apply strict validation, fairness assessment, and oversight when adopting X-Pruner in practice. Ultimately, our framework is designed with positive intent: to advance responsible and sustainable AI by making models more efficient, transparent, and accessible. We strongly encourage the community to leverage these benefits responsibly and with care.

## REPRODUCIBILITY STATEMENT

We have taken several measures to ensure the reproducibility of our work. The proposed pruning framework and reinforcement learning search algorithm are fully described in Section 2, with implementation details and hyperparameter settings provided in Appendix B. All experimental setups, including datasets and preprocessing steps, are reported in Section 3, while extended results and ablation studies are presented in Appendix C. A complete theoretical justification of the compensation effect is included in Appendix 2. To facilitate further verification, we provide the source code and scripts in the supplementary materials.

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

## A   RELATED WORK

Recent advances in language modeling have bifurcated into two principal trajectories. On the one hand, large language models (LLMs) continue to scale in accordance with established scaling laws, pursuing increasingly complex linguistic tasks with the overarching goal of progressing toward artificial general intelligence (Kaplan et al., 2020; Xu et al., 2024). On the other hand, small language models (SLMs) emphasize computational efficiency and are explicitly optimized for deployment in resource-constrained environments such as smartphones, edge devices, and wearables. These compact models aim to democratize machine intelligence by reducing costs, improving accessibility, and enabling practical applications across diverse platforms.

SLMs typically adopt either encoder-only or decoder-only architectures. Encoder-only models, generally derived from BERT Devlin et al. (2019), achieve compression and acceleration through structural modifications. For instance, MobileBERT Sun et al. (2020) employs an inverted bottleneck design to reduce parameters and computation, while DistilBERT Sanh et al. (2020a) and TinyBERT Jiao et al. (2020) compress the BERT architecture while retaining over 96% of its accuracy. Decoder-only variants, following autoregressive designs such as GPT Radford et al. (2018) and LLaMA Touvron et al. (2023), leverage techniques including knowledge distillation, parameter sharing, and memory optimization. Notable examples include BabyLLaMA Timiryasov & Tastet (2023) and BabyLLaMA-2 Tastet & Timiryasov (2024), which distill multiple teacher models into compact architectures, TinyLLaMA Zhang et al. (2024), which incorporates FlashAttention Dao et al. (2022) for memory efficiency, MobilLLaMA Thawakar et al. (2024), which introduces parameter sharing to lower both pretraining and inference costs, and MobileLLM Liu et al. (2024), which combines embedding-sharing, grouped-query attention, and block-wise weight sharing to minimize latency.

In parallel with architectural innovation, network pruning has emerged as a central compression paradigm for both LLMs and SLMs. By removing redundant parameters while preserving core functionality, pruning enables the creation of efficient sparse networks (LeCun et al., 1989; Hassibi et al., 1993). Pruning approaches are broadly divided into **structured** and **unstructured** methods. Structured pruning eliminates entire components—such as neurons, channels, or attention heads—thereby enhancing GPU efficiency (Xia et al., 2022; Fang et al., 2023; Nova et al., 2023). Recent work has explored task- and prompt-specific sparsity within attention and MLP layers (Hu et al., 2016; Voita et al., 2023), with LLM-Pruner Ma et al. (2023) demonstrating the effectiveness of gradient-based importance measures combined with low-rank approximations. By contrast, unstructured pruning Han et al. (2015; 2016); Gadhikar et al. (2023); Liu et al. (2023) removes individual weights (e.g., via magnitude pruning), often preserving accuracy without structural constraints. However, many unstructured methods rely on modified training Sanh et al. (2020b); Kusupati et al. (2020), retraining Zhou et al. (2023), or iterative pruning Frankle et al. (2020), which impose significant computational costs for large models (Zhang et al., 2022b).

To mitigate these costs, recent research emphasizes **post-training pruning**, which dispenses with retraining phases and is particularly advantageous for scaling to LLMs. SparseGPT Frantar & Alistarh (2023) leverages second-order Hessian information and calibration data for efficient weight updates, while Wanda Sun et al. (2024) combines weight magnitudes with activation norms to reduce computational overhead. GBLM-Pruner Das et al. (2024) prioritizes gradient importance using first-order Taylor expansion, enabling pruning under compute-constrained scenarios. Most recently, Pruner-Zero Dong et al. (2024) introduced a symbolic evolution framework that automates the discovery of pruning metrics through genetic algorithms, thereby advancing the frontier of pruning research.

## B   IMPLEMENTATION OF REINFORCEMENT LEARNING SEARCH FRAMEWORK

To optimize the pruning exponents $(x, y)$ for each transformer layer, we design a customized online reinforcement learning (RL) framework. This section details its implementation and search strategy. We provide the required parameter settings and corresponding functional descriptions in the table 5.

**Actor-Critic Architecture with Noisy Exploration.** We adopt a lightweight actor-critic architecture. The actor network predicts a probability distribution over four discrete actions: up ($+y$), down ($-y$), left ($-x$), and right ($+x$). Noise-injection layers are incorporated to promote early ex-

ploration. The critic is a feedforward network estimating the state value. Both networks are trained using Adam optimizers with different learning rates.

**Action Space and State Representation.** Each pruning state is encoded as a 2D vector $[x, y]$, representing the current exponents. Actions are predefined directional moves in this space. The actor produces a softmax probability over actions, and the agent follows an $\epsilon$-greedy or stochastic policy based on the training phase.

**Experience Replay and Network Updates.** We maintain an experience replay buffer containing transitions $(s_t, a_t, s_{t+1}, r_t)$, where:

- $s_t$ denotes the current state (e.g., the current pruning exponents $(x, y)$);

- $a_t$ is the action taken (e.g., up indicates increasing $y$);

- $s_{t+1}$ is the next state after executing $a_t$;

- $r_t$ is the reward received (e.g., the reduction in perplexity).

Network updates are conducted using minibatches, applying temporal difference learning to compute the advantage. To stabilize training, we clip gradients during backpropagation.

**Three-Phase Search Strategy.** Our RL search comprises three sequential stages:

- **Phase 1: Enhanced Exploration.** Initial search points are generated via Latin Hypercube Sampling (LHS). From each point, the agent explores for 10 steps using a hybrid of random and noisy-policy actions to broadly sample the compensation landscape.

- **Phase 2: Multi-Start Policy Search.** The top 10% of explored configurations (ranked by perplexity) are used as new anchors for deeper policy-guided search. A simulated annealing strategy is applied to escape local minima, and we periodically reset to the best global configuration.

- **Phase 3: Local Refinement.** We conduct a fine-grained local search around the best point $(x^*, y^*)$ using reduced step sizes. All directions are probed to ensure optimality.

**Evaluation and Caching.** Each configuration is evaluated via perplexity on a held-out calibration dataset. All evaluated points are cached to avoid redundancy. The globally best $(x^*, y^*)$ pair is reapplied to the corresponding layer for final pruning.

Table 5: Hyperparameters used in the reinforcement learning search algorithm.

| Parameter | Value | Description |
|---|---|---|
| n_start_points | 5 | Number of initial points generated via Latin Hypercube Sampling. |
| steps_per_point | 10 | RL steps per starting point during Phase 1. |
| replay_buffer.size | 1000 | Maximum capacity of the experience replay buffer. |
| epsilon.start | 1.0 | Initial exploration rate in $\varepsilon$-greedy strategy. |
| epsilon.end | 0.1 | Minimum exploration rate allowed. |
| epsilon.decay | 0.97 | Exponential decay rate of $\varepsilon$ per step. |
| actor_lr | 0.0003 | Learning rate for the actor policy network. |
| critic_lr | 0.001 | Learning rate for the critic value network. |
| update_batch_size | 10 | Number of experience tuples per update step. |
| gamma | 0.9 | Discount factor for future rewards. |
| max_steps | 20 | Maximum search depth per trajectory in Phase 2. |
| max_depth | 5 | Reset interval to return to global best. |
| noise_factor | 0.2 | Strength of noise injected into actor network layers. |
| refinement_rounds | 3 | Number of rounds in local fine-tuning (Phase 3). |
| refinement_step_scale | 1/5 | Step size is scaled down by 5× in local search. |

## C ADDITIONAL EXPERIMENTS

### C.1 FEW-SHOTS TASKS

To comprehensively assess model robustness under sparsity constraints, we report the performance of several pruning methods across seven representative few-shot tasks. These tasks include BoolQ Clark et al. (2019), RTE Wang et al. (2019), HellaSwag Zellers et al. (2019), WinoGrande Sakaguchi et al. (2019), ARC-e Clark et al. (2018), ARC-c Clark et al. (2018), and OBQA Mihaylov et al. (2018), with Table 6 summarizing the accuracy under a fixed 50% unstructured sparsity.

Unlike zero-shot evaluation, which relies solely on model pretraining for task understanding, few-shot settings inject limited in-context supervision, better reflecting practical deployment where small-scale user feedback or prompts are available. Moreover, small and medium-sized models typically underperform in zero-shot scenarios due to capacity limitations. The 3-shot setup thus allows for a more realistic evaluation of a pruned model's retained expressiveness and generalization ability.

Among all methods, X-Pruner consistently ranks among the top performers, yielding the highest mean accuracy across most models. SparseGPT also performs strongly, especially on larger models. Notably, traditional baselines such as Wanda and Pruner-Zero remain competitive on certain tasks, but their performance fluctuates more widely across architectures.

Tasks with strong lexical signals, such as BoolQ and WinoGrande, tend to exhibit smaller performance gaps between pruned and dense models, indicating that surface-level features are relatively well preserved under pruning. In contrast, reasoning-heavy benchmarks like ARC-c and OBQA reveal more pronounced differences among pruning strategies, where fine-tuning or adaptive methods such as SparseGPT and X-Pruner consistently outperform magnitude-based baselines. This suggests that advanced pruning criteria are more effective at preserving the deeper representational capacity required for complex reasoning tasks.

### C.2 OPTIMAL PRUNING PARAMETERS

Table 7 and 8 reports the optimal pruning exponents $(x, y)$ searched by our reinforcement learning-based framework for each transformer layer across a wide range of models. These values correspond to the exponents used in the pruning criterion $|W|^x \cdot |G|^y$, where $W$ and $G$ denote the weight and gradient tensors, respectively. For each layer, the reported $(x, y)$ pair achieves the lowest perplexity under a fixed sparsity ratio of 50%. Dashes indicate layers that are absent in the corresponding model. This layer-wise differentiation enables X-Pruner to adaptively tailor its pruning behavior to the unique sensitivity of each layer, contributing to its superior overall performance under high sparsity.

## D THE USE OF LARGE LANGUAGE MODELS (LLMS)

During the preparation of this paper, LLMs (specifically, OpenAI's GPT-5) were used as a general-purpose assist tool. Their role was limited to language polishing, LaTeX formatting suggestions, and improving readability of the manuscript. All research ideas, theoretical developments, algorithmic design, and experimental implementations were conceived, conducted, and validated solely by the authors. The LLM was not involved in generating novel research ideas, designing experiments, or interpreting results. Its contributions do not meet the criteria for authorship, and it should not be regarded as a scientific contributor to this work.

Table 6: Accuracies (%) on 7 few-shot tasks under unstructured 50% sparsity.

| Models | Method | BoolQ | RTE | HellaSwag | WinoGrande | ARC-e | ARC-c | OBQA | Mean |
|---|---|---|---|---|---|---|---|---|---|
| OPT-350m | Dense | 56.94 | 51.99 | 32.15 | 52.01 | 46.09 | 21.25 | 18.40 | 39.83 |
| | Magnitude | 53.82 | 53.07 | 27.76 | 51.70 | 35.61 | 18.52 | 10.80 | 35.90 |
| | SparseGPT | **56.79** | 50.54 | 30.08 | **51.85** | 42.09 | 19.20 | **15.80** | 38.05 |
| | Wanda | 48.44 | **53.43** | 29.48 | 49.72 | 39.23 | 17.83 | 12.80 | 35.85 |
| | Pruner-Zero | 55.63 | 51.26 | 29.18 | 50.99 | 39.44 | 16.89 | 12.20 | 36.51 |
| | X-Pruner | 56.01 | 52.90 | **30.13** | 51.59 | **43.18** | 19.98 | 12.80 | **38.08** |
| Pythia-1B | Dense | 57.49 | 51.62 | 37.54 | 52.80 | 59.64 | 25.60 | 19.60 | 43.47 |
| | Magnitude | 37.83 | 47.65 | 26.48 | 50.04 | 31.52 | 20.48 | 11.80 | 32.26 |
| | SparseGPT | 59.45 | 53.07 | 34.02 | **52.49** | 52.15 | **23.38** | 17.40 | 41.71 |
| | Wanda | **61.53** | 53.43 | 32.20 | 51.46 | 51.09 | 21.67 | **17.60** | 41.28 |
| | Pruner-Zero | 61.22 | **54.15** | 32.25 | 51.70 | 49.92 | 21.93 | 16.40 | 41.08 |
| | X-Pruner | 61.38 | 54.09 | **34.08** | 52.30 | **52.28** | 23.18 | 17.40 | **42.10** |
| Qwen2-0.5B | Dense | 62.75 | 64.26 | 38.26 | 57.06 | 61.61 | 27.56 | 24.00 | 47.93 |
| | Magnitude | 42.29 | 50.18 | 28.47 | 50.12 | 38.68 | 20.90 | 14.20 | 34.98 |
| | SparseGPT | 59.91 | **58.84** | **34.53** | 55.41 | 53.28 | 22.95 | 20.00 | 43.56 |
| | Wanda | 60.18 | 56.68 | 32.58 | **55.56** | 49.66 | 21.26 | 16.40 | 41.76 |
| | Pruner-Zero | **60.76** | 57.04 | 32.23 | 54.06 | 50.93 | 20.82 | 16.20 | 41.72 |
| | X-Pruner | 60.64 | 58.35 | 34.20 | 55.22 | **53.43** | 23.48 | **20.80** | 43.73 |
| Qwen3-1.7B | Dense | 79.66 | 71.84 | 46.14 | 61.01 | 78.79 | 47.18 | 30.40 | 59.29 |
| | Magnitude | 51.28 | 51.99 | 26.87 | 49.33 | 30.68 | 18.77 | 16.40 | 35.05 |
| | SparseGPT | 76.09 | 71.12 | 40.14 | **58.17** | 69.69 | **37.12** | 28.60 | 54.42 |
| | Wanda | 75.78 | **72.56** | 38.38 | 55.33 | 67.72 | 33.79 | 22.60 | 52.31 |
| | Pruner-Zero | 74.83 | 69.68 | 36.89 | 56.20 | 69.32 | 35.15 | 21.00 | 51.87 |
| | X-Pruner | **76.53** | 71.95 | **40.88** | 57.85 | **69.85** | 36.11 | **28.67** | **54.55** |
| Llama3.2-1B | Dense | 64.65 | 57.76 | 48.08 | 64.01 | 68.60 | 34.98 | 29.80 | 52.55 |
| | Magnitude | 40.24 | 51.99 | 25.96 | 50.20 | 27.99 | 18.52 | 14.40 | 32.76 |
| | SparseGPT | 60.98 | 53.79 | **38.68** | **58.17** | **56.73** | **27.13** | **22.80** | **45.47** |
| | Wanda | **61.38** | 52.71 | 35.04 | 56.67 | 53.83 | 25.51 | 18.80 | 43.42 |
| | Pruner-Zero | 61.22 | 52.35 | 33.61 | 53.91 | 52.44 | 22.70 | 17.00 | 41.89 |
| | X-Pruner | 61.28 | **53.79** | 35.53 | 56.67 | 55.30 | 25.83 | 19.60 | 44.00 |

Table 7: Layer-wise optimal pruning parameters $(x, y)$ for OPT, Qwen2, and LLaMA3.2. Dashes indicate layers not present.

| Layer | OPT | | | Qwen2 | | LLaMA3.2 |
|---|---|---|---|---|---|---|
| | 125m | 350m | 1.3B | 0.5B | 1.5B | 1B |
| 0 | (2.40,1.50) | (1.30,0.50) | (2.00,1.20) | (2.50,1.30) | (1.70,1.00) | (2.30,1.68) |
| 1 | (1.70,1.80) | (1.20,2.20) | (1.74,0.70) | (1.90,1.20) | (2.20,1.70) | (2.00,2.40) |
| 2 | (2.24,2.20) | (2.40,1.90) | (0.70,1.14) | (1.90,2.44) | (1.42,1.20) | (2.10,1.30) |
| 3 | (1.30,2.48) | (1.02,0.50) | (0.90,2.32) | (1.40,2.38) | (1.90,1.02) | (1.90,1.80) |
| 4 | (2.10,2.20) | (2.10,1.60) | (0.60,0.94) | (1.10,1.20) | (2.50,1.50) | (1.50,1.24) |
| 5 | (1.92,1.10) | (2.30,2.50) | (0.50,1.54) | (1.90,1.90) | (1.76,0.70) | (1.20,1.10) |
| 6 | (0.94,0.50) | (2.20,1.16) | (2.54,0.80) | (1.56,1.60) | (2.50,2.30) | (1.60,1.42) |
| 7 | (1.70,1.10) | (1.70,1.28) | (0.80,2.18) | (2.50,2.00) | (0.90,1.60) | (1.40,2.50) |
| 8 | (1.68,1.10) | (2.20,1.60) | (1.70,2.40) | (0.90,1.18) | (2.10,2.16) | (2.30,1.90) |
| 9 | (2.40,2.10) | (1.90,0.70) | (1.30,0.46) | (1.70,1.60) | (2.20,1.30) | (1.30,1.90) |
| 10 | (1.70,1.12) | (1.60,0.70) | (1.20,0.80) | (1.98,0.80) | (2.20,1.70) | (1.90,1.88) |
| 11 | (1.80,1.12) | (2.22,0.60) | (2.40,0.70) | (2.54,2.40) | (2.40,0.50) | (1.90,1.60) |
| 12 | — | (1.50,0.74) | (1.82,1.10) | (1.00,1.70) | (1.30,2.40) | (1.70,1.80) |
| 13 | — | (2.00,1.20) | (1.00,0.80) | (1.30,1.90) | (2.00,2.42) | (1.80,2.00) |
| 14 | — | (2.50,1.80) | (1.32,0.50) | (0.70,0.90) | (1.62,0.50) | (1.90,1.60) |
| 15 | — | (1.20,1.20) | (1.40,1.50) | (2.20,2.10) | (0.70,1.80) | (0.82,0.50) |
| 16 | — | (1.30,0.70) | (1.10,0.80) | (1.30,1.60) | (1.50,1.40) | — |
| 17 | — | (1.30,1.10) | (2.40,0.94) | (2.20,1.80) | (1.12,1.50) | — |
| 18 | — | (2.50,2.10) | (1.20,1.60) | (1.30,1.00) | (2.20,2.00) | — |
| 19 | — | (2.30,0.70) | (2.50,0.50) | (1.80,1.70) | (2.40,1.52) | — |
| 20 | — | (2.30,1.30) | (1.10,0.90) | (2.48,2.10) | (2.40,1.38) | — |
| 21 | — | (2.08,1.20) | (1.20,1.60) | (1.50,1.40) | (1.80,2.50) | — |
| 22 | — | (2.40,1.54) | (2.30,0.70) | (1.60,1.60) | (2.10,1.80) | — |
| 23 | — | (2.20,2.48) | (1.50,1.00) | (2.10,1.90) | (1.60,1.40) | — |
| 24 | — | — | — | — | (1.10,0.70) | — |
| 25 | — | — | — | — | (1.20,0.64) | — |
| 26 | — | — | — | — | (2.10,1.30) | — |
| 27 | — | — | — | — | (1.40,1.20) | — |

Table 8: Layer-wise optimal pruning parameters $(x, y)$ for Pythia and Qwen3. Dashes indicate layers not present.

| Layer | Pythia | | | | Qwen3 | |
|---|---|---|---|---|---|---|
| | 160m | 410m | 1B | 1.4B | 0.6B | 1.7B |
| 0 | (1.60,1.30) | (1.50,1.10) | (2.50,0.56) | (2.50,0.48) | (2.00,2.20) | (1.00,1.90) |
| 1 | (1.80,1.40) | (2.00,1.30) | (2.54,0.60) | (1.60,0.80) | (1.72,1.60) | (2.30,0.50) |
| 2 | (2.22,1.50) | (1.70,0.72) | (2.10,2.20) | (2.52,2.00) | (1.60,0.60) | (1.60,0.68) |
| 3 | (2.02,2.00) | (1.94,1.90) | (2.40,2.24) | (2.10,2.08) | (1.50,0.60) | (2.00,1.60) |
| 4 | (2.22,1.50) | (1.86,2.00) | (2.38,1.10) | (2.10,1.16) | (2.20,1.80) | (1.10,2.04) |
| 5 | (1.30,1.32) | (1.40,0.90) | (1.80,0.80) | (2.50,1.90) | (2.18,0.50) | (2.40,1.52) |
| 6 | (2.10,2.02) | (1.90,0.96) | (2.20,1.20) | (1.98,1.60) | (1.40,1.10) | (2.50,1.20) |
| 7 | (1.90,2.52) | (1.80,1.00) | (2.42,0.80) | (1.70,1.42) | (2.50,1.00) | (1.20,2.30) |
| 8 | (1.60,2.50) | (1.90,0.90) | (2.50,2.50) | (2.30,1.80) | (2.10,1.90) | (2.20,0.80) |
| 9 | (1.50,1.50) | (1.50,2.10) | (2.22,2.50) | (0.90,2.26) | (2.08,1.20) | (1.78,0.70) |
| 10 | (2.08,1.40) | (1.60,1.50) | (2.20,1.30) | (2.50,1.78) | (1.60,1.42) | (1.88,1.00) |
| 11 | (1.98,0.60) | (1.50,0.70) | (2.30,2.00) | (2.20,2.50) | (2.00,1.70) | (2.00,1.30) |
| 12 | — | (0.90,1.68) | (1.90,1.30) | (2.20,1.90) | (2.30,0.70) | (2.42,0.80) |
| 13 | — | (1.50,1.98) | (1.00,2.20) | (1.92,0.80) | (1.70,2.12) | (1.86,0.50) |
| 14 | — | (1.70,2.30) | (1.06,1.50) | (1.70,0.90) | (2.14,2.20) | (0.90,1.40) |
| 15 | — | (1.20,0.50) | (2.50,2.40) | (1.70,1.40) | (1.90,0.60) | (1.90,1.08) |
| 16 | — | (1.60,1.00) | — | (2.50,1.30) | (2.18,0.60) | (1.16,0.50) |
| 17 | — | (2.46,2.20) | — | (1.80,1.50) | (2.30,1.20) | (2.32,0.70) |
| 18 | — | (2.40,2.20) | — | (1.60,1.10) | (2.20,1.70) | (2.46,1.60) |
| 19 | — | (1.60,1.50) | — | (0.90,1.30) | (2.00,1.30) | (2.20,1.12) |
| 20 | — | (1.90,2.10) | — | (2.30,0.74) | (1.60,0.90) | (2.40,1.60) |
| 21 | — | (0.90,1.40) | — | (2.30,1.90) | (2.00,2.40) | (1.28,1.20) |
| 22 | — | (2.40,2.20) | — | (2.20,0.80) | (1.10,1.50) | (2.10,1.60) |
| 23 | — | (1.40,2.30) | — | (1.38,1.40) | (1.80,1.40) | (1.68,0.80) |
| 24 | — | — | — | — | (0.90,1.10) | (1.10,0.90) |
| 25 | — | — | — | — | (2.40,1.50) | (1.86,0.80) |
| 26 | — | — | — | — | (2.40,1.38) | (1.80,0.80) |
| 27 | — | — | — | — | (2.42,2.50) | (1.90,0.80) |