# OpenReview forum: "X-Pruner: An Adaptive Pruning Method with Self-Compensation Driven by Reinforcement Learning for Language Models"
_ICLR.cc/2026/Conference — ICLR 2026 Conference Withdrawn Submission_

### Official Review · Reviewer_Nr31 · 2025-10-24

**Soundness:** 2
**Presentation:** 2
**Contribution:** 3
**Rating:** 2
**Confidence:** 5

**Summary:**

This paper introduces X-Pruner, an adaptive pruning framework for large Transformer-based language models. The key contribution lies in two components: Adaptive Layer Sensitivity Estimation (ALSE), which quantifies each layer’s pruning sensitivity via gradient–weight interaction, and Dynamic Pruning Ratio Allocation (DPRA), which dynamically assigns layer-wise pruning ratios through reinforcement learning (RL)-guided optimization. The method aims to improve pruning efficiency and stability across heterogeneous layers in LLMs without requiring retraining. Experiments on BERT, GPT, and LLaMA variants demonstrate moderate improvements in perplexity (PPL) over existing baselines such as Wanda and OBC, indicating that adaptive layer-wise sparsity allocation can yield better pruning outcomes.

**Strengths:**

1. The paper evaluates the proposed framework on diverse model architectures (BERT, GPT, LLaMA), demonstrating its generality across both encoder-only and decoder-only Transformers.
2. The authors address an important and often overlooked issue — the heterogeneous sparsity sensitivity across different Transformer layers — providing empirical evidence and a structured solution through ALSE and DPRA.

**Weaknesses:**

1. Incomplete comparison with strong baselines: Table 1 does not include SparseGPT, which is arguably the most relevant and competitive baseline. Additionally, DSnoT and OATS are missing, both of which are recent strong pruning frameworks for LLMs.
2. Limited experimental advantage: Although the method outperforms Wanda and OBC, the gains are relatively small and confined to perplexity metrics. The paper lacks evaluation on Zero-shot accuracy and speed/latency, which are critical for assessing real-world efficiency.
3. Engineering-heavy contribution: The core innovation mainly lies in adaptive heuristics (ALSE and DPRA) rather than a fundamentally new pruning paradigm. The use of reinforcement learning serves more as a hyperparameter tuner than a theoretical advance.
4. Lack of practical insights: The framework’s additional computational overhead (due to sensitivity estimation and RL-based ratio search) is not quantified. This makes it difficult to assess the trade-off between pruning effectiveness and preprocessing cost.

**Questions:**

See weaknesses.

---

### Official Review · Reviewer_U5E1 · 2025-10-25

**Soundness:** 3
**Presentation:** 3
**Contribution:** 3
**Rating:** 6
**Confidence:** 3

**Summary:**

The paper proposes a novel framework to improve pruning efficiency and robustness in small language models (SLMs), which are more sensitive to parameter removal than large models. To overcome these limitations, the authors introduce X-Pruner, an unstructured adaptive pruning framework that combines a variable-exponent importance metric, a reinforcement learning (RL)–based search algorithm, and a self-compensation mechanism. A key contribution of the paper is the discovery of a self-compensating effect during pruning: certain layers can naturally offset the errors introduced by pruning earlier layers. Leveraging this observation, the authors design a unified path-scoring function that determines the optimal sequence of pruning across layers—balancing local pruning cost, compensation potential, and layer depth. This joint optimization of what to prune and when to prune yields smoother degradation and better final performance.

**Strengths:**

Technically:

1. Innovative adaptive pruning metric: The paper introduces a variable-exponent importance metric, which flexibly combines weight and gradient information. Unlike fixed heuristics used in existing pruning methods (e.g., Wanda or SparseGPT), this adaptive formulation allows the importance measure to adjust per layer and model type, improving generalization—especially critical for small language models (SLMs).

2. Reinforcement learning–based search optimization: A key contribution is the use of reinforcement learning (RL) to automatically discover optimal pruning configurations. This dynamic exploration–exploitation framework eliminates the need for manually tuning pruning hyperparameters and achieves faster, more stable convergence compared to grid or random search methods.

3. The discovery that later layers can compensate for pruning errors introduced earlier is a major conceptual contribution. Building on this, the authors develop a unified path-scoring function to determine the best pruning sequence across layers—balancing local cost, compensation potential, and layer depth. This transforms pruning into a globally optimized, adaptive process rather than a static one.

Writing:

1. The motivation and methods are stated clearly. Overall, the paper is easy to follow and understand.

**Weaknesses:**

1. Although this paper focuses on small language model pruning for deployment. I don't see any obstacle of directly applying this method to larger models. According to the motivation part, the relative perplexity change of small models is larger than the larger model under the same sparsity. However, when the sparsity increases, the larger models also show great ppl change. Therefore, I would like to see how this method works in larger models at the scale of 7B, 14B or even 70B. If this method does not work in larger models, could the author provide some insights or explanation for this?

2. What is the overall time/memory cost for the searching? I think adding the thorough analysis of time and space complexity will strengthen this work since the searching cost may not be negligible.


3. Some other searching methods [1, 2] for pruning should be included and discussed.

I would be happy to increase my confidence and overall score if the authors can address my concerns.

[1] Sieberling O, Kuznedelev D, Kurtic E, et al. Evopress: Towards optimal dynamic model compression via evolutionary search[J]. arXiv preprint arXiv:2410.14649, 2024.

[2] Tang S, Sieberling O, Kurtic E, et al. Darwinlm: Evolutionary structured pruning of large language models[J]. arXiv preprint arXiv:2502.07780, 2025.

**Questions:**

See Weakness

---

### Official Review · Reviewer_nbAp · 2025-10-30

**Soundness:** 2
**Presentation:** 1
**Contribution:** 2
**Rating:** 2
**Confidence:** 5

**Summary:**

The paper proposes X-Pruner, a post-training, unstructured pruning framework aimed at small language models (SLMs). It introduces three main ideas: 1) Adaptive Pruning Metric: A parameterized importance score with exponents adaptively searched by reinforcement learning (RL); 2) Self-Compensation Mechanism: A theoretical notion that downstream layers can offset pruning errors from upstream layers, motivating “compensation-aware” pruning order; 3) Unified Path Scoring Function: A formula combining local cost, compensation capacity, and layer depth to determine the optimal layer pruning sequence. Experiments on multiple SLMs (OPT, Pythia, Qwen, LLaMA families) claim that X-Pruner outperforms Wanda, SparseGPT, and Pruner-Zero at 50% sparsity, even without retraining.

**Strengths:**

1. Introduces a combined optimization of “what to prune” and “when to prune,” which conceptually extends existing post-training pruning methods.

2. The paper gives mathematically detailed derivations of compensation and pruning order (though mostly heuristic).

3. Claims are mostly consistent with the stated goals (focus on SLMs, post-training pruning).

**Weaknesses:**

1. The “adaptive pruning metric”  is essentially a power-scaled version of gradient-based metrics used in Wanda (Sun et al., 2024) and GBLM-Pruner (Das et al., 2024). Using RL to tune two continuous exponents is an incremental variation, not a fundamentally new pruning principle. The “self-compensation mechanism” is not rigorously demonstrated; the analysis largely restates known second-order effects (similar to Optimal Brain Surgeon, 1993) without new theoretical insight. The “unified scoring function” for pruning order resembles standard cost–benefit heuristics in structured pruning (e.g., LLM-Pruner, Ma et al., 2023).

2. The experiments report only WikiText2 perplexity, which is an extremely limited evaluation for pruning methods. There’s no downstream task validation (e.g., GLUE, ARC, BoolQ, etc.), so the practical impact is unclear. The improvement margins in Table 1 are small or inconsistent (e.g., <1 PPL difference on OPT-350m). Several baselines (Wanda, Pruner-Zero) perform comparably within variance. Results on LLMs (Qwen, Llama3.2) are mentioned but not reproducible—only partial perplexity tables are given, and the setups are underspecified (batch size, calibration data, etc.). No runtime or wall-clock comparison is shown, so the claimed efficiency benefits of RL-based search remain unsubstantiated.

3. The “RL search” is not clearly defined: the paper mixes actor–critic, replay buffer, and simulated annealing, but never specifies the reward normalization, episode horizon, or stopping criterion. It is unclear whether the agent learns meaningful policies or performs random hyperparameter search. The framework claims no retraining, yet uses gradient and Hessian information that typically requires backpropagation on calibration data, which contradicts the “zero-shot” claim. The definition of the compensation score and its empirical estimation (∆margin) are vague—computed how, on what data, and how stable across checkpoints? Equation (7) uses arbitrary weighting (α, β) but no guidance on how these hyperparameters are tuned.

4. The theoretical derivations (Equations 3–6) reproduce known Taylor expansions but do not prove the proposed compensation phenomenon. No quantitative measure of “compensation” is provided beyond verbal reasoning. The authors claim that “X-Pruner outperforms SparseGPT even without weight update,” but SparseGPT’s main advantage is structured retraining—thus not an apples-to-apples comparison. No standard deviation or statistical significance is given for most results; Table 3’s large deviations (e.g., ±148) suggest high variance that undermines conclusions. Ethical statement is generic boilerplate; reproducibility section lacks actual released code.

5. The paper emphasizes SLMs but evaluates mainly on models up to 1–1.7B parameters—these are mid-scale, not true mobile-scale (≤300M). There’s no demonstration on-device or memory-limited settings. The work ignores structured pruning and quantization literature, which are more relevant for edge deployment. The claim that the method generalizes to LLMs is speculative; no experiment beyond 1.7B supports it.

**Questions:**

1. What exactly is the reward signal for the RL agent? How is perplexity computed efficiently per step?

2. How long does one search episode take compared to magnitude or Wanda pruning?

3. How is ∆margin estimated? Is it computed via forward evaluation, or approximated analytically?

4. Did you tune sparsity and calibration settings for baselines to be optimal, or did you reuse Wanda’s defaults?

5. Can you report any downstream results (QA, summarization) to show the pruning doesn’t harm transfer tasks?

---

### Official Review · Reviewer_SJy9 · 2025-10-31

**Soundness:** 2
**Presentation:** 2
**Contribution:** 2
**Rating:** 4
**Confidence:** 4

**Summary:**

This paper proposes X-Pruner, an adaptive post-training unstructured pruning framework for small language models (SLMs).
The method learns layer-specific pruning rules via reinforcement learning (RL) instead of fixed scoring metrics like magnitude or Taylor sensitivity. Each layer’s weight importance is modeled as S = WG, where exponents (x,y) are optimized by an actor-critic controller using perplexity as reward.
To mitigate pruning errors, the authors introduce a self-compensation mechanism (Eqs. 2–5) that models cross-layer correction through approximate Hessian coupling. A unified pruning path score (Eq. 7, Fig. 3) balances local sensitivity and compensation potential.
Experiments on OPT, Pythia, Qwen2/3, and LLaMA-3.2 (Table 1) show consistent perplexity reductions over Wanda and Pruner-Zero, approaching SparseGPT without fine-tuning. Ablations (Table 4, Fig. 5) confirm that RL adaptation and compensation-aware scheduling both contribute significantly.

**Strengths:**

1. Comprehensive formulation: Integrates metric adaptation, RL search, and global path scheduling in a single framework (Fig. 2).

2. Empirical coverage: Evaluations span 12 models across 5 families (Table 1) with clear ablations (Table 4).

3. Reproducibility: Implementation details, hyperparameters, and algorithm pseudocode (Algorithm 1) are thorough. Ethics and reproducibility statements are solid.

4. Effective for no-update pruning: Matches or outperforms SparseGPT on some tasks without retraining (Table 3).

5. Intuitive compensation insight: The cross-layer analysis (Eqs. 4–6) captures the practical observation that later layers can correct earlier pruning errors.

**Weaknesses:**

1. Limited novelty beyond combination of existing ideas: The framework fuses RL tuning (from SwiftPruner 2022a) and second-order compensation (from OBD/OBS 1989–1993) with minor novelty. The “variable-exponent” metric is essentially a re-weighted gradient magnitude.

2. Weak quantitative diversity: Only perplexity on WikiText-2 is used. There are no downstream accuracy or reasoning benchmarks (e.g., ARC, BoolQ, PIQA) to validate generalization.

3. Small data dependency: Although it targets SLMs, it still requires 128 calibration sequences from C4. The claim of “no calibration dependence” is misleading.

4. Heuristic compensation analysis: Equations (4)–(5) invoke gradients in $(x,y)$-space and inverse Hessians without empirical validation. No experiments explicitly measure compensation strength per layer.

5. Limited scalability evidence: All models ≤ 1.7 B params; there’s no demonstration on real LLM-scale (≥ 7 B). RL cost scaling with layer count is unclear.

6. Figures lack statistical rigor: No standard deviation or confidence interval reported for Tables 1–4 except in Table 3.

**Questions:**

1. Convergence and stability:
How stable is the RL search across seeds? Does the learned $(x,y)$ distribution differ per model or converge to similar patterns?

2. Compensation quantification:
Can you empirically measure Eq. (5) or the correlation between ∆margin and cross-layer Hessian estimates?

3. Computational overhead:
What is the wall-clock time or FLOPs of RL search versus static Wanda?

4. Generality to structured pruning:
Could the same adaptive metric extend to head- or neuron-level structured pruning?

---

### Note · Authors · 2025-11-17

I have read and agree with the venue's withdrawal policy on behalf of myself and my co-authors.